# Membranes for Osmotic Power Generation by Reverse Electrodialysis

**DOI:** 10.3390/membranes13020164

**Published:** 2023-01-28

**Authors:** Md. Mushfequr Rahman

**Affiliations:** Helmholtz-Zentrum Hereon, Institute of Membrane Research, Max-Planck-Straße 1, 21502 Geesthacht, Germany; mushfequr.rahman@hereon.de; Tel.: +49-4152872446

**Keywords:** reverse electrodialysis, osmotic power, blue energy, porous membrane, ion selective membrane

## Abstract

In recent years, the utilization of the selective ion transport through porous membranes for osmotic power generation (blue energy) has received a lot of attention. The principal of power generation using the porous membranes is same as that of conventional reverse electrodialysis (RED), but nonporous ion exchange membranes are conventionally used for RED. The ion transport mechanisms through the porous and nonporous membranes are considerably different. Unlike the conventional nonporous membranes, the ion transport through the porous membranes is largely dictated by the principles of nanofluidics. This owes to the fact that the osmotic power generation via selective ion transport through porous membranes is often referred to as nanofluidic reverse electrodialysis (NRED) or nanopore-based power generation (NPG). While RED using nonporous membranes has already been implemented on a pilot-plant scale, the progress of NRED/NPG has so far been limited in the development of small-scale, novel, porous membrane materials. The aim of this review is to provide an overview of the membrane design concepts of nanofluidic porous membranes for NPG/NRED. A brief description of material design concepts of conventional nonporous membranes for RED is provided as well.

## 1. Introduction

Climate change, the energy crisis and the scarcity of fresh water are the pressing global grand challenges of the 21st century. In spite of the tremendous growth of sustainable energy generation technology in the last decades, these challenges remain interconnected with each other. Still, the global energy generation landscape is largely dependent on fossil fuels. The salinity gradient energy generation technology reverse electrodialysis (RED) is a promising candidate for reducing the current dependency of the energy portfolio on fossil fuels. The utilization of RED is mostly focused on using the river water as a dilute solution and seawater as a concentrated solution due to the availability of the resources, especially at the estuaries. The maximum global potential of osmotic power from the seawater and river water mixing is estimated to be 2.6 TW. Considering the technological issues, around 980 GW or 0.98 TW of power is extractable from the mixing of seawater and river water [1,2,3]. The low salinity of river water is a major obstacle for the utilization of this resource, which decreases the achievable gross power from an RED stack and hinders the implementation of RED for large-scale power production. A potential strategy for overcoming this technological limitation is using saline water as the low-concentration feed solution and concentrated brine as the high-concentration feed solution of RED [4,5]. In this regard, energy production by RED using the waste brine stream of the seawater desalination plants has received a lot of attention [6]. According to a survey [7] in 2018, there are 15,906 operational desalination plants located in 177 countries, producing 95.37 million m^3^/day of fresh water from different sources, e.g., seawater, brackish water, etc. Reverse osmosis (RO), as the dominant desalination technology in the global market, is used in 13,446 plants [7]. The conventional surface water sources (e.g., rivers, lakes) are not enough to meet the rising global demand of fresh water. Seawater desalination by RO (SWRO) is envisaged to become the key solution for mitigating the future water demand. The specific energy consumption of SWRO (2.5–4.0 kWhm^−3^) is 8 to 10 times higher than that required for the production of fresh water from conventional sources (0.2–0.4 kWhm^−3^) [8]. The specific energy consumption of SWRO is dependent on several factors, such as the type of SWRO configuration (e.g., single-pass SWRO, two-pass SWRO, etc.), the site-specific factors (e.g., source and product water quality specification), the efficiency of energy recovery devices, etc. [9]. The integration of RED with SWRO is a green technology for compensating a part of the energy consumed in SWRO [4,6,10,11]. The potential of using the municipal wastewater as a feed stream for RED has also received a lot of attention. The brine of seawater desalination plants can be mixed with the municipal wastewater stream to extract energy by RED. Approximately 18.5 GW of energy can be extracted by utilizing the municipal wastewater that is discharged in the sea [2]. The first ever lifecycle assessment of energy recovery by RED from the waste streams of a two-pass SWRO plant was published in 2020 [6]. This study shows that the mixture of the retentate of second-pass SWRO and the effluent of a wastewater treatment plant can be used as the low-concentration feed stream of RED. The dilution of the concentrate of the second-pass SWRO increases the extractable thermodynamic energy. However, it is possible to adopt such configuration only if a wastewater treatment plant is present next to the SWRO plant [6]. Apart from integrating RED with SWRO and wastewater remediation technology, other unique applications of RED are under consideration as well [12]. An off-grid, decentralized sanitation strategy for recovering energy from human urine is suggested, using a hybrid RED and membrane distillation technology [13]. RED is usually considered in an open-loop configuration. A closed-loop configuration termed as an RED heat engine has been proposed, where the high-concentration and the low-concentration feed streams of RED are regenerated using a heat engine as a post-treatment process [12]. Daniilidis et al. [14] have thoroughly investigated the experimentally obtainable power density of RED using NaCl solutions ranging from 0.01 M to 5 M to mimic river water, seawater, concentrated brine, etc. and have stated that “*The results suggest that there is no single way to improve the performance of a RED system for all concentrations. Improvements are therefore subject to the specific priorities of the application and the salt concentration levels used*” [14]. Future research on RED membranes should be directed toward adopting material design strategies that aim to solve the limitations of the technology, taking into account the salinity of the feed solutions for a particular application of RED. The lack of the development of specifically tailored ion exchange membranes for a particular application of RED has been pointed out as one of the major limitations in the progress of RED technology [5,15,16]. In the last decade, the development of membranes for RED has received a lot of attention. While conventionally nonporous anion exchange and cation exchange membranes were used for RED, at present, ion selective porous membranes are under the limelight [17]. The initial studies of porous membranes started with investigations of ion transport through a single pore and calculating the maximum output power under a given salinity gradient. For example, 20 pW of power is generated by a single boron nitride nanotube using a 1000-fold salinity gradient of KCl [18]. The linear extrapolation of the obtained output power for predicting the performance of multi-porous membranes with a pore density of 10^8^–10^10^ cm^−2^ suggests that the utilization of porous membranes may surpass the maximum output power density of conventional nonporous membranes by several orders of magnitude. This has attracted such strong attention that the application of porous membranes for osmotic power generation has turned into an independent research domain. The ion transport mechanism of the porous membranes is largely governed by the principles of nanofluidics. To distinguish from the conventional nonporous RED membranes, this field is often referred to as nanofluidic reverse electrodialysis (NRED) [1,19] or nanopore-based power generation (NPG) [20,21,22]. The development of porous membranes for osmotic power generation has progressed a long way beyond such single-pore demonstration. A large number of materials and fabrication methods have been reported for NRED/NPG. However, the challenges of using porous membranes for real applications have also been realized. The fabrication of a real membrane of a large area with a sufficient number of pores per unit area that can achieve the excellent predicted power density by the single-porous material is a gigantic task, if at all possible. At present, it is well established that the power density of the membrane does not increase linearly with the pore density of the membranes up to 10^8^–10^10^ cm^−2^. The power density of the membrane starts to drop as the pore density exceeds a certain value, which is in the range of 10^4^ cm^−2^ [20]. This is a consequence of the concentration polarization which is inherent to every membrane-based separation with aqueous feed solutions. For membranes with a high pore density, the concentration polarization reduces the ion selectivity and ion conductance, which leads to a decrease in power density. Due to this inherent limitation of porous membranes, their superiority over the conventional nonporous membranes has been questioned [20,21]. This review reports the membrane material development strategies for osmotic power generation by RED and NRED/NPG. As the principles of RED have been thoroughly covered in several reviews [3,16,23,24], they have not been discussed thoroughly here. For the convenience of the readers, a short description is provided (Section 2). The concepts of material design and the structure property relationship of conventional nonporous ion exchange membranes are briefly covered (Section 3). The membrane fabrication strategies of the porous membranes for NRED/NPG are the major focus of this review paper (Section 4).

## 2. Principles of Reverse Electrodialysis

Gibbs free energy is released upon the mixing of solutions with different salinities. RED can be regarded as a method for converting the free energy of mixing into electrical energy. For the basic research of RED membrane material development, two chamber cells are used. The chambers contain two electrodes and salt solutions of different salinities separated by an ion selective semipermeable membrane (Figure 1a). This setup is termed as a half-cell system [17]. Both porous and nonporous membranes used as the semipermeable barrier generate either a positive or negative surface charge in a hydrated state. Osmotic pressure builds up across the membrane owing to the different salinities of the salt solutions in the two chambers. The surface charge of the membranes attracts counter ions and repels the co-ions. Compared to the co-ions, the counter ions migrate from the high-concentration side to the low-concentration side much more easily. A net charge migration across the membrane occurs owing to the selective ion transport which creates a potential across the membrane. To maintain the electroneutrality of the system, a redox reaction occurs at the electrodes, which converts the net charge migration across the membrane into an electron flow to the external circuit. In this way, the Gibbs free energy of mixing (which forces the transport of ions from the high-concentration solution towards the low-concentration solution) is converted to electrical energy [17,25]. The power harvested from the salinity gradient in this way is output to an electronic load resistor, *R_L_*. If *I* is the measured current, the electric power density, *P_R_*, can be calculated from the equation *P_R_ = I^2^ × R_L_*. Typically, *I* is measured for a range of *R_L_*, and the respective *P_R_* is calculated. The *P_R_* reaches a maximum peak value when *R_L_* is equal to the internal resistance of the source [26]. The peak value, termed as the maximum output power density, is generally used as the parameter to benchmark the performance of different membranes.

For large-scale power generation, a full-scale system is used, which is often termed as an RED stack (Figure 1b). An RED stack consists of an alternating series of anion and cation exchange membranes separated by spacers to create solution chambers. As the solutions are pumped through the chambers, the anions and cations selectively permeate through the respective membranes from the high-concentration solution toward the low-concentration solution. In this way, the positive ions and the negative ions are forced to move in opposite directions, which creates positively and negatively charged poles in the RED stack. Two electrodes are placed at each end of the RED stack, where redox reactions take place to convert the ionic flux into electricity. The chemical potential difference between the two solutions with different salinities gives rise to a voltage across each of the ion exchange membranes. The sum of the voltage across each of the RED membranes of the stack (i.e., the open-circuit voltage) and the internal resistance of the RED stack determines the gross electrical power [24]. The net power output of RED is the gross power minus the power consumed to pump the solutions through the RED stack. The pumping of the feed solution consumes a substantial amount of energy. Veerman et al. [27] reported that the energy required for pumping the feed solutions is around 25% of the total power generated by a seawater/river water RED stack containing 50 cells. Considering the hydrodynamic loss, it is important to take care of several aspects of the RED stack design in order to maximize the net output power. The intermembrane distance and feed water flow rate have a direct influence on the gross power of an RED stack [28]. The conventional concept was to use non-conductive spacers (such as a spacer made of polyethylene terephthalate) to separate the anion exchange and cation exchange membranes of the RED stack. These spacers reduce the available surface area of the membranes by covering a part of the membranes. This effect (widely termed as the spacer shadow effect) reduces the output power. Two strategies have been proposed to eliminate the spacer shadow effect: (i) using ion conductive spacers [29] and (ii) using profiled membranes [30,31], i.e., membranes with ridges at the surface to separate the next membrane. Concentration polarization occurs due to the difference in the ionic mobility in the solution phase and the membrane phase. As a result, a diffusion boundary layer builds up at the membrane surface, which contributes to the increase in the stack resistance, i.e., the reduction in output power. While, like every other membrane-based technology with aqueous feed solutions, concentration polarization cannot be eliminated in RED, it is essential to minimize this effect by optimizing the stack hydrodynamics [32,33]. Another important aspect is to ensure the uniform distribution of the solution at the membrane surface by mitigating fouling. The flow of feed solutions locally decreases at the regions where fouling accumulates and the effective membrane area decreases. Thus, it is also important to figure out the cleaning intervals of an RED stack to maximize the gross power [34].

## 3. Conventional Nonporous Membranes

A large number of RED studies have been performed using commercially available ion exchange membranes which have originally been designed for other applications, e.g., Neosepta^®^ (designed for electrodialysis), Fumasep^®^ (designed for diffusion dialysis), Ralex^®^ (designed for electrodialysis), Selemion^®^ (designed for diffusion dialysis), etc. [14,33,35,36]. Due to the completely different driving force and operating condition, the physical and electrochemical requirements of the membrane materials are not similar for all applications [16]. The driving force of ion transport through the RED membranes stems from the salinity gradient of the feed streams. The RED membranes are expected to have a high ion selectivity and low resistance. The selectivity is the ability of the membrane to allow for the permeation of the counter ions while rejecting the co-ions. The resistance is the tendency of the membrane to prevent the fast permeation of ions, i.e., resist the passage of ionic current [37]. Thus, the resistance decreases with the thickness of the membrane. A high selectivity and low resistance are naturally counteracting. Ion exchange membranes are generally composed of immobilized ionic functional groups in a polymeric substrate. The cation exchange membranes usually contain carboxylic or sulfonic functional groups, while the anion exchange membranes contain ammonium functional groups [16]. The parameter ion exchange capacity (IEC) is widely used as a metric of the content of the ionic functional groups of the RED membranes [38]. The electrostatic exclusion of ions (i.e., Donnan exclusion) is an important phenomenon in determining the selectivity of the membranes. Although the selectivity of the membrane is expected to increase with the increase in the IEC due to the higher Donnan exclusion, this is not always the case due to the influence of other factors. Due to the presence of the ionic functional groups, the membrane uptakes water and swells in a hydrated state. As the membrane swells, the ion permeation through the membrane increases, i.e., the resistance of the membrane decreases. However, along with the counter ions, the co-ions also start to permeate through the swelled membrane, which means that the membrane starts to lose the selectivity [37]. Moreover, both the extent of the swelling of the membrane and the Donnan exclusion are related to the salinity of the aqueous solution. The osmotic deswelling of the membrane leads to a lower water uptake (i.e., swelling) of the membrane as the salinity of the aqueous solution increases. The Donnan exclusion of co-ions also decreases with increasing salt concentration [37]. Thus, even the ion exchange membranes with the optimum combination of selectivity and resistance for an RED stack with low-salinity feed solutions will not lead to the best performance of another RED stack using feed solutions of a substantially higher salinity. It is important to find a balance between the counteracting selectivity and resistance of the RED membranes, taking into account the salinity of the feed solutions of an RED stack. In conjunction with the requirement of finding a balance between these counteracting features, the adverse effect of the uphill transport of the divalent ions (from the low-concentration solution towards the high-concentration solution) on gross power adds further difficulty in the optimum design of the RED membranes [24]. At present, it is well known that the key to controlling these properties of the membranes is controlling the IEC and the swelling of the membrane in a hydrated state. The state-of-the-art concept of the structure property relationship is to correlate the selectivity and the resistance of the membranes with the metric of fixed charge density (FCD), which is the ration of the IEC and water uptake (which is the measure of swelling) of the membrane [37,39,40]. FCD does not take into account the distribution of ionic functional groups in the membrane. In terms of the macroscopic distribution of ionic functional groups, the ion exchange membranes are broadly classified into two categories: homogenous membranes and heterogeneous membranes. Homogeneous membranes are polymer membranes where the functional groups are directly attached on the polymer backbone. In the heterogeneous membranes, ion exchange beads are hot-pressed in a polymer matrix. The properties of both homogeneous and heterogeneous commercially available membranes have been investigated in the context of RED [14,33,35,36,37,40,41]. In general, heterogeneous membranes have a higher resistance compared to homogeneous membranes [35]. The first tailor-made RED membranes reported by Kitty Nijmeijer and co-workers in 2012 are homogeneous crosslinked membranes [39]. Blends of polyepichlorohydrin and polyacrylonitrile are used as the polymer matrix, and 1,4-diazabicyclo(2.2.2)octane (DABCO) is used as the crosslinker for the preparation of these membranes. The IEC and swelling degree of these membranes increase simultaneously with the decrease in the polyacrylonitrile content. In these membranes, polyacrylonitrile is used as a blend partner of polyepichlorohydrin to improve the mechanical properties. Free standing membranes up to 33 µm in thickness are prepared in this technique, which are threefold thinner than the commercially available ion exchange membranes [39]. This casting solution is also used to prepare double-layer membranes. A crosslinked selective layer that is 5–8 µm in thickness is obtained by the spin coating of the casting solution on top of an anodized alumina oxide support [42]. The reaction between DABCO and the chlorine moieties of polyepicholorohydrin introduces charged moieties at the crosslinked junctions [39,43,44]. The anion exchange membrane obtained by this technique can also be post-modified with polyethyleneimine and glutaraldehyde to increase the anion selectivity [45]. It is generally established that, for a crosslinked polymer membrane, the degree of swelling decreases with the increase in the degree of crosslinking. Thus, the FCD of a crosslinked membrane can be tuned by varying the degree of crosslinking and IEC. There are several metrics expressing the degree of crosslinking of a crosslinked polymer network: the concentration of elastically active chains *ν_el_/V*_0,_ the crosslink density *μ_el_/V*_0_, the cycle rank density, *ξ/V*_0_, and the averaged molecular weight of the polymer chains between the crosslinked points, *M_C_*, where *ν_el_* is the number of elastically active chains, *μ_el_* is the number of crosslinks, *ξ* is the number of independent circuits in the network (cycle rank) and *V*_0_ is the volume of the dry polymer network [46,47,48]. To the best of my knowledge, the FCD, selectivity and resistance of the membrane have never been investigated for a crosslinked membrane where any of these metrics expressing the degree of crosslinking was known. In other words, the current structure property correlation concept of the RED membranes completely ignores the influence of the topology of the crosslinked polymer membrane on the ion permeation. When a crosslinked membrane swells in a hydrated state, molecular-level cavities are formed due to the stretching of the chains between the crosslinked point. The dimensions of these cavities are not only determined by the extent of the swelling but also by the chain topology. A polymer network with a broad distribution of *M_C_* will have a broad size distribution of the cavities. In my opinion, diffusion will play a relatively stronger role in the selectivity for feed solutions with a higher salinity due to the weaker electrostatic exclusion of the co-ion. Thus, the influence of the chain topology on the selectivity of the membrane will be higher for feed solutions with a higher salinity. The degree of the swelling of the membrane itself also decreases with the increase in the salinity (osmotic deswelling effect). It is a difficult task to establish a systematic relationship between the permeation behavior and the topology of a crosslinked polymer network.

Another approach for preparing RED membranes is to introduce ionic functional groups on a non-crosslinked hydrophobic polymer backbone, e.g., sulfonated polyether ether ketone [40], quaternary ammonium functionalized poly(2,6-dimethyl-1,4-phenylene oxide) [37], etc. The presence of pendant ionic functional groups tends to swell the membrane, while the hydrophobic polymer backbone tends to prevent the swelling. Therefore, for the membranes prepared from these polymers, a compromise of IEC is required to tune the swelling of the membrane. Too high of an IEC may lead to a complete loss of the ion selective nature of these membranes due to excessive swelling. A strategy for controlling the swelling in spite of having a sufficient IEC is designing tailor-made copolymers with both ionic repeating units and hydrophobic non-ionic repeating units (e.g., fluorinated monomers). In this way, the FCD of the membrane can be controlled by changing the ration of the ionic and non-ionic monomers of the copolymer. In this regard, tailor-made sulfonated poly(phenyl-alkane)s has been reported for osmotic power generation. The polymers are synthesized by a one-step Friedel–Crafts reaction between 6,6′-dimethoxy-3,3,3′,3′-tetramethyl-1,1′-spirobiindane and two commercial benzaldehyde derivatives. Disodium 4-formylbenzene-1,3-disulfonate is used as the sulfonated monomer, while triflourobenzaldehyde is used as the hydrophobic moiety. For a 50-fold salinity gradient of NaCl (i.e., using artificial river and seawater), an 8.23–9.58 Wm^−2^ maximum output power density is obtained from the free-standing membranes produced from this class of polymer [49].

## 4. Emerging Porous Membranes

### 4.1. Selective Ion Transport through Porous Membranes

The development of ion selective porous membrane materials for osmotic power generation has been a vibrant field of research in the last decade. The potential of using porous membranes for the selective transport of ions was realized due to the ion-current rectification behavior of the nanoporous system. It refers to the deviation of the empirically observed current–voltage curve from the ohmic behavior. Although the concentration and pH of the electrolytes in contact with both pore openings are the same when a voltage difference is applied across the nanoporous material, higher or lower magnitudes of measured currents are observed at negative potentials compared to those at positive potentials. In other words, an asymmetric current–voltage curve is observed instead of a linear one (i.e., ohmic behavior), owing to the selective permeation of either anion or cation through the nanoporous system [50,51]. Typically, asymmetric nanopores (e.g., cone shaped) with a pore radius smaller than or similar to the Debye screening length exhibit ion–current rectification. There is an intense research effort to understand the underlying molecular-level mechanism behind this interesting phenomenon [50,51]. For osmotic power generation, the driving force of ion transport does not come from applied voltage. However, it is a common trend to study the current–voltage curve (i.e., the I–V curve) of a membrane at an equal concentration and pH of the electrolytes under applied voltage, along with that under a salinity gradient of electrolytes. It is helpful to understand the ion transport behavior of the membrane in this way. For such a fundamental study, membranes with a single pore are the most attractive choice, as it is possible to directly observe the nanoscale transport behavior through one pore without having to average the effect of multiple pores. In the single-pore membrane, the geometry of the pore can be well controlled, and the influence of pore geometry on ion transport can be studied. Moreover, the change in the concentration of the low-concentration and high-concentration chambers of the measurement cell due to the transport through the single channel is negligible. Hence, the driving force of ion transport, i.e., the salinity gradient during the whole set of the current–voltage curve measurement, can be kept constant very easily [52]. The ion selectivity of a porous membrane is largely dictated by the pore size and surface charge density. When a porous membrane is immersed in an aqueous solution containing salts, the charge density induces the formation of an electric double layer (EDL) (Figure 2a). However, unlike nonporous membranes, the EDL does not form only at the surface of the membrane but also inside the pores of the porous membranes. The EDL inside the pores consists of the Stern layer and the diffuse layer. The combined thickness of the Stern layer and the diffuse layer is often termed as the length of the EDL and is conventionally referred to as the Debye length. The ionic strength of the salt solution has a strong influence on the Debye length, which may vary between tens of nanometers and less than one nanometer. The Debye length (*κ*^−1^) is expressed using the following equation:(1)$$\kappa^{-1}=\left(\frac{\varepsilon k_{B}T}{e^{2}\sum n_{i}^{2}c_{i}}\right)$$
where *ε* is the permittivity of the solvent, *k_B_* is the Boltzmann’s constant, *T* is the absolute temperature, *e* is the charge of the electron, *n_i_* is the charge of the ion and *c_i_* is the bulk concentration of the ion. If the radius of the pore is equal to or smaller than the Debye length (*κ*^−1^), the EDL will overlap [1]. For a charged surface with EDL, the electric potential is highest at the surface. With the increase in the distance from the surface, the potential exponentially decreases to zero. There is an equal probability of finding co-ions and counter ions at a distance from the surface where the potential is zero. However, in a charged nanopore with a radius equal to or smaller than the Debye length, the potential does not reach zero because of the overlapping of the EDL. In other words, if the pore radius of a charged pore equals the Debye length, there will be a higher number of counter ions inside the pore compared to the co-ions [1]. Therefore, due to the driving force of the salinity gradient between the concentrated and dilute salt solution, the counter ions will transport more easily compared to the co-ions (Figure 2b). In short, for porous membranes, the selectivity is a result of electrostatic interactions between the pore wall and mobile ions. From the above discussion, it is clear that pores with a longer length and narrower diameter will lead to a higher ion selectivity, as it offers a higher charged surface area and confinement to attract the counter ion and repel the co-ion. However, at the same time, the ion permeability through the membrane will decrease with the increase in the pore length, i.e., the resistance of the membrane will increase [53]. The resistance of the membrane decreases with the length of the pore, but the maximum output power and the energy conversion efficiency drop sharply for membranes with excessively short pores. Apart from the loss of ion selectivity, the negative impact of concentration polarization becomes rather strong in cases of pores with an extremely short length. As a result, the transmembrane salinity gradient cannot be well maintained for membranes with an extremely short pore length [54]. It is evident that finding the optimum combination of the pore diameter, pore length and surface charge density is of utmost importance to maximizing the output power of NRED/NPG. In this regard, several empirical [52,55] and simulation [54,56,57] studies have been devoted to thoroughly examining the impact of the length, diameter and surface charge density of a nanoscale pore on the ion permeation, the ion selectivity and, eventually, the potential output power.

The transport of liquids through nanoscale pores differs fundamentally from the macroscopic transport. The classical theories of fluid mechanics assume a no-slip boundary condition at the solid–liquid interface. According to these conventional theories, when water flows through a porous medium, the relative velocity of water in contact with the solid surface is zero. In other words, the water molecules in contact with the solid surface are completely immobilized due to the infinite solid–liquid friction. Due to this assumption of a no-slip boundary condition, the conventional theories fail to explain the water flow behavior through nanoscale pores. For a nanoscale pore with a hydrophobic surface, the friction at the solid–liquid interface is rather low, i.e., the hydrodynamic slip is high. As a result, the water flows faster through such nanoscale pores than what is predicted by the Hagen–Poiseuille law [58,59]. In NRED/NPG, the ion transport through the pores does not occur only due to diffusion and electromigration. Ions also transport through the pores due to water flow. Diffusoosmosis is expected to play a major role in the transfer of the net ionic charge across the nanofluidic porous membrane, thereby generating a large power density [18,57]. The term diffusoosmosis refers to the flow of fluid induced by the interaction between a solid surface and solute owing to the imposed concentration gradient [60]. Therefore, it is important to consider the influence of the slip on the ion transport through the pores [57]. The solid–liquid friction at the interface is generally expressed in terms of slip length. It is defined as the distance from the surface at which the linear extrapolation of the velocity profile vanishes. A small slip length is associated with high solid–liquid friction at the interface [58,61]. While the slip length of water and the solid surface crucially depend on the wettability of the solid surface, the interfacial hydrodynamic slip is related with other surface properties as well, e.g., roughness [53,57]. As the slip length increases, the diffusoosmotic flow of ions through the pores increases along with the increase in water flow. Hence, an increase in slip length increases the maximum power output due to the higher ion permeability through the pore. Continuum fluid dynamics simulation suggests that a threefold higher maximum output power density can be achieved by the pores having a 100 nm slip length compared to having no slip [57]. It is worth mentioning here that, while in hydrophobic surfaces, slip lengths in the range of few tens of nanometers have been reported, in hydrophilic surfaces, the slip length is in terms of sub-nanometers [58]. Recent simulation studies have revealed that not only the slip length of the interior surface of the pores but also the slip length of the exterior surface have an impact on the ion current migration through the membrane. Hence, designing membranes with a hydrophobic exterior surface and a charged interior pore surface can be a promising approach to improving the maximum output power density [53].

### 4.2. Track-Etched Polymer Membranes with 1D Pores

The track etching of polymer films is an ideal technique for generating both single-porous and multi-porous membranes. This pore generation technique involves a bombardment of high-speed heavy ions at the surface of the polymer films, followed by chemical treatment. Poly(ethylene terephthalate) and polyimide are the most popular polymers used to prepare the track-etched porous membranes. The control over the number of ions bombarded at the polymer film surface allows for the fabrication of membranes with the desired pore density. In this technique, membranes with a pore density as low as one pore per square centimeter can be prepared. Due to the chemical etching, the pores of both poly(ethylene terephthalate) and polyimide are decorated with carboxylic groups, which induces a negative charge to the pores. It also allows for the possibility of tuning the ion transport through the pore by changing the pH of the electrolyte solutions. The shape of the pore can be controlled by varying the condition of chemical etching. The fast penetration of the etchant along the ion bombarded track leads to cylindrical pores. To generate cone-shaped pores, a strong chemical etchant is introduced from one side of the film, while a so-called stopping medium for neutralizing the chemical etchant is used at the other side of the film [51]. Track-etched polyimide membranes that are 12 µm in thickness and have a cone-shaped single pore have been used by Guo et al. [52] to demonstrate the potential of osmotic power generation. A total of 1 mM of a KCl aqueous solution was used at the tip side of the cone-shaped pore, while the concentration of the KCl aqueous solution at the base side of the pore varied between 100 and 1000 mM. A maximum power of 26 pW was obtained at a 1000-fold concentration gradient, along with a pH of 5.6. As it is possible to produce track-etched membranes with pore densities in the range of 10^8^–10^10^ cm^−2^, Gu et al. [52] estimated that such nanoporous membranes will have a power density of 2–260 mW (i.e., 20–260 Wm^−2^), which is substantially higher than the power density achievable by the commercially available nonporous ion exchange membrane. However, as already mentioned in Section 1, at present, it is well established that the pore density and maximum power output of the membrane do not follow a linear relationship. The geometry of the single-porous track-etched membrane has a substantial impact on the power density. Recently, it has been demonstrated that a single bullet-shaped pore on a 12 µm-thick poy (ethylene terephthalate) film produces 80 pW of maximum power, which surpasses the maximum power of pores with cylindrical, funnel and conical geometry (Figure 3) [62].

The track-etched poly(ethylene terephthalate) pores are capable of absorbing polyelectrolytes. This provides an easy way to functionalize the pore wall and tune the interior surface charge by the deposition of polyelectrolytes. As the track-etched poly(ethylene terephthalate) pores contain carboxylic groups in the pore wall, the selective ion transport through the pore can be tailored by the deposition of a positively charged polyelectrolyte at the interior pore wall, e.g., poly(allylamine hydrochloride) [63]. A track-etched conical pore on poly(ethylene terephthalate) after functionalization by poly-L-lysine produces 120 pW of power for a 500-fold KCl concentration gradient [64]. Another method is to use the layer-by-layer assembly of two oppositely charged polyelectrolytes, which provides a better control over the surface charge at the interior of the pore. Several combinations of oppositely charged polyelectrolytes have been successfully used to tailor the surface pore charge and thereby the ion current rectification of the track-etched poly(ethylene terephthalate) pores, e.g., poly(allylamine hydrochloride) and poly(styrene sulfonate) [65], polyethylenimine and chondroitin-4-sulfate [66], poly-L-lysine and poly(styrene sulfonate) [67], etc. A fivefold increase in output power is obtained by functionalizing a track-etched single-porous poly(ethylene terephthalate) membrane by the layer-by-layer assembly of chitosan and poly(acrylic acid). For a 1000-fold salinity gradient of NaCl at pH 7.6, the maximum power output of the polyelectrolyte-functionalized single pore is 25pW, while that of the pristine single pore is 5.2 pW [68]. The fabrication of both anion-selective and cation-selective membranes has been proposed by the in situ synthesis of a highly charged gel directly inside the pores of track-etched poly(ethylene terephthalate) membranes. In this regard, anion- and cation-selective gels were synthesized inside the cylindrical and conical pores using (3-acrylamidopropyl)-trimethylammonium and 3 sulfopropyl acrylate, respectively [69]. The gels dictated the ion transport property through the membrane undermining the influence of the cylindrical and conical pore geometry. With a stack of four membranes (two anion exchanges and two cation exchanges), a 0.37 Wm^−2^ power density was achieved at pH 7 for a 1000-fold salinity gradient of NaCl [69]. The post-modification of the exterior surface of the track-etched poly(ethylene terephthalate) membrane with polyaniline allows for the pH-dependent modulation of ion transport over a broad pH range. The ion transport through the polyaniline-coated track-etched poly(ethylene terephthalate) becomes redox stimuli-dependent owing to the electrochemically active nature of polyaniline. Upon coating the exterior surface of the tip side of a bullet-shaped pore by polyaniline, 15 pW of power is generated by the pore at pH 3.5 for a 1000-fold concentration gradient of KCl [70].

### 4.3. Porous Membranes with Atomic- and Molecular-Scale Thickness

Atomically thick porous materials are attractive for NPG/NRED, as the resistance of the membranes scales proportionally to the thickness. In this class of membrane, it is crucial to control the thickness of the membrane down to a single atomic layer parallel to generating the pores of a desired size at a high pore density [71]. Meanwhile, the mechanical stability of the atomically thick porous layer is a major concern as well. The utilization of a focused electron beam of a transmission electron microscope for puncturing a hole in graphene brought the advent of atomically thin porous membranes [72,73]. While this method offers good control over the pore size, it is not suitable for scaling up the membranes. Therefore, alternative methods for the generation of pores in graphene have been explored as well. The two-step atom-by-atom nucleation and growth of pores involving the generation of defects by argon ion bombardment followed by selectively etching the edges of the defect by a diffuse electron beam have been introduced [74]. A further modification of this method involving the generation of defects by gallium ion bombardment followed by oxidative etching for generating pores using acidic potassium permanganate has been reported as well. In this method, pores with 0.4 ± 0.24 nm diameters and pore densities of 10^12^ cm^−2^ can be generated. The pores generated by this method show a modest selectivity of K^+^ over Cl^−^ owing to the presence of functional groups at the pore edges, e.g., hydroxyl, carboxyl, quinone, ketone, etc. [75]. The porous molybodenum disulfide (MoS_2_) [76] and hexagonal boron nitride (hBN) [77] monolayers have been used to demonstrate the potential power generation by NPG/NRED. Typically, the monolayers MoS_2_ or hBN are grown on top of a nonporous substrate (e.g., sapphire) by chemical vapor deposition and then transferred on top of a porous silicon nitride (SiNx) layer. Finally, pores are generated on the monolayer. The electron beam drilling using a transmission electron microscope [78] is the conventional way to generate pores on the MoS_2_ layer, but the nanoscale pore on the MoS_2_ layer can also be generated by an electrochemical reaction [79]. In this method, the composite of nonporous MoS_2_ and porous SiNx is placed in a two-chamber cell containing aqueous buffer (1 M KCl, pH 7.4) solutions, and a transmembrane potential is generated using two electrodes. At a voltage higher than the oxidation potential of MoS_2_ in the aqueous medium, single atoms or unit cells of MoS_2_ are successively removed from the MoS_2_ lattice to generate a pore [79]. Tip-controlled local break-down using an atomic force microscope has been used to prepare nanoscale pores on a boron nitride monolayer on top of a porous silicon nitride layer [77]. The biggest advantage of this AFM-based pore fabrication technique is that it allows for the generation of a multiparous hBN layer with a controlled spatial positioning of the pore. In other words, this technique allows for the generation of pores with a well-defined pore-to-pore distance. The most attractive feature of both MoS_2_ and hBN is their high negative charge at the surface, which makes a large contribution to the selective cation transport through the pore. The charge density at the pore opening of the MoS_2_ monolayer varies with the size of the pore. A change in the surface pore density from −0.024 cm^−2^ to −0.088 cm^−2^ was reported for an increase in the pore diameter from 2 nm to 25 nm. A 15 nm pore showed a selectivity of 0.4 from 1M KCl to 1 m KCl at pH 5, while a 5 nm pore showed perfect cation selectivity at this operating condition. The Debye length of a 1 mM KCl solution is 10 nm, which means the EDL overlaps inside the pore at this pore size range. The molecular dynamics simulation shows that the maximum power decreases with the increase in MoS_2_ layers. Using both empirical and simulation studies, it has been demonstrated that a high surface charge and an atomically thin layer of MoS_2_ would lead to efficient osmotic power generation [76]. Yazda et al. [77] reported a −0.25 cm^−2^ surface charge for porous hBN membranes. A 7 nm single-porous hBN membrane showed a cation selectivity of 0.53 from 1 M KCl to 1 m KCl at pH = pKa. This study reported the osmotic current and osmotic potential of single-porous hBN membranes with a pore diameter in the range of 3–16 nm. As the pore diameter crossed 10 nm, the osmotic potential and osmotic current across the membrane dropped sharply, which suggests a loss of the ion selectivity of the membrane. As the Debye length of 1 M KCl is 10 nm, the overlap of EDL is expected for all membranes with a pore diameter of 3–16 nm. The loss of ion selectivity suggests that, as the membrane is very thin, although the dilute solution side tends to induce overlapped EDL inside the pore, both counter ions and co-ions enter the pore from the concentrated solution side and prevent the formation of overlapped EDL inside the entire pore. The study of Yazda et al. [77] is not only limited to the single-porous membranes. They have also studied multi-porous membranes with a pore-to-pore spacing in the range from 100 nm to 1 µm. The maximum power does not increase linearly with the number of pores. A maximum power density is observed for the critical pore-to-pore distance of 500 nm. For lower pore-to-pore distances, the power density drops significantly. A maximum power density of 15 Wm^−2^ was obtained from a 1000-fold salinity gradient of KCl. The fabrication of ultrathin porous layers is not only limited to the top-down methods discussed above. Several bottom-up strategies have been reported, which inherently have a better potential in scaling up the fabrication technique compared to the top-down techniques. The oxidative polymerization of 2,3,6,7,10,11-hexaminotriphenylene (HATP) at an organic/water interface is a scalable bottom-up approach to fabricating multi-porous graphene sheets containing –NH_2_ groups at the pore edges [80]. This technique also generates sub-nanometer pores with a pore density of 10^12^ cm^−2^. The atomically thin porous membranes prepared in this method produce a 35 Wm^−2^ maximum power density for a 100-fold salinity gradient of KCl at ambient pH [80]. Crosslinking a core–rim structure polycyclic aromatic hydrocarbon monomer hexa(2,2′-dipyridylamino)hexabenzocoronene is another bottom-up approach to preparing molecularly thin porous carbon membranes [81]. The polycyclic aromatic hydrocarbon is composed of six flexible dipyridylamino groups (i.e., the rim) covalently linked to a hexabenzocoronene core. Owing to the amphiphilic nature, the polycyclic aromatic compound forms a monolayer on a water surface. The monolayer is transferred on a copper substrate, followed by annealing at 550 °C to form the ultrathin layer. The pores of the ultrathin layer have a diameter of 3.6 nm, while the pore density is 1.2 × 10^10^ cm^−2^. A free-standing membrane with a 1 µm diameter prepared in this method produces a maximum power density of 67 Wm^−2^ for a 50-fold NaCl salinity gradient. A centimeter-scale 2 nm-thick carbon layer prepared by this method is mounted on a porous polycarbonate track-etched membrane without the formation of cracks [81]. The well-ordered pores of the covalent organic framework (COF) monolayer also demonstrate ultrahigh ion conductivity [82]. A maximum output power of 135.8 Wm^−2^ is obtained from the mixing of NaCl solutions of a 50-fold salinity gradient using a metal tetraphenylporphyrin COF (MTPP-COF) monolayer containing a square-shaped nanopore array. The fabrication of the imine-linked conjugated skeleton of the MTPP-COF monolayer consists of a Schiff-base condensation reaction between a zinc tetraphenylporphyrin monomer with an amine functional group and 2,5-dihydroxyterephthaladehyde. Usually, the cation selective membranes produce less power in electrolytes containing divalent cations compared to that in NaCl. However, the MTTP-COF monolayer membrane produces 317.5 Wm^−2^ and 267.7 Wm^−2^ of power in electrolytes at a 50-fold salinity difference of MgCl_2_ and CaCl_2_, respectively. As the seawater contains a considerable number of divalent ions, this exceptional result obtained from the MTPP-COF monolayer is the beginning of a new era of cation selective membranes for NRED/NPG [82].

### 4.4. Nanofluidic Membranes Having 2D Pores

In 2D nanofluidic membranes for osmotic power generation, the ion transport occurs through the interstitial gaps between the planar 2D materials, e.g., graphene, Mxene, molybdenum disulfide, etc. [22]. An exfoliation–reconstruction approach (Figure 4a) is commonly employed to prepare such membranes with massive arrays of 2D nanofluidic channels [83,84]. Upon the restacking of the exfoliated nanosheets, a layered structure with tunable interlayer spacing is formed. Usually, the lateral size of the nanosheets is highly polydisperse. The reconstructed stack of nanosheets usually forms highly tortuous interconnected nanofluidic channels of just a few nanometers. In such membranes, the interlamellar gap of the nanosheets has to be comparable with the Debye length in order to ensure the overlap of the electric double layer for selective ion transport (Figure 4b) [83,84]. The nanosheets of graphene oxide, the most popular building block of such 2D nanofluidic membranes, are typically ~1 nm-thick, while the lateral dimensions are of tens of nanometers. A graphene oxide nanosheet is decorated with epoxy, phenyl hydroxyl and carboxylic groups, which renders a negative surface charge in a hydrated state. The vacuum filtration of a dispersion with exfoliated graphene oxide nanosheets is a common way to prepare the graphene oxide stack [85]. Owing to the high aspect ratio, the graphene oxide nanosheets stack horizontally, which forms a paper-like layered structure. As the lateral dimensions of the graphene oxide nanosheets are much smaller than the dimensions of the graphene oxide paper, the 2D channels between a pair of nanosheets are highly interconnected. It is worth mentioning that stacking defects definitely occur during the reconstruction of the exfoliated graphene oxide sheets. However, such stacking defects can result in a nonselective microscopic void in graphene oxide paper only if the defects percolate along the whole cross-section. [84] It is convenient to tune the surface charge of the exfoliated nanosheets by functionalization before the reconstruction step to avoid the challenges of the chemical post-modification of the layered stack. While the pristine graphene oxide nanosheets are negatively charged in a hydrated state, a positive charge can be induced by converting the carboxylic groups to alkyl-substituted imidazolium groups. The reaction steps involve the dispersion of graphene oxide in N,N-dimethylformamide and the addition of 1-(3-Dimethylaminopropyl)-3-ethylcarbodiimide in the dispersion. Upon the addition of 1-aminopropyl-3-methylimidazolium bromide in the dispersion, positively charged graphene oxide is formed, which can be easily separated from the dispersion by centrifugation. The zetapotential of the graphene oxide colloids shifts from −42 mV to +55 mV, owing to this chemical post-modification. A 0.77 Wm^−2^ maximum power density is obtained using a negatively charged graphene oxide membrane (surface charge density −123 mC m^−2^) and a positively charged graphene oxide membrane (surface charge density +147 mC m^−2^) in a three-chamber cell for a 50-fold salinity gradient of NaCl [86].

Apart from graphene oxide, selective ion transport has been reported through the interlayer space of other 2D nanosheets as well. A stack of boron nitride nanosheets have around 5-9 Å interlayer spacing, which is large enough to accommodate two to three layers of water molecules. The small two-dimensional nanochannels of boron nitride allow for the selective permeation of cation owing to their negative surface charge in a hydrated state [87]. The chemical vapor deposition or vapor deposition polymerization of melamine leads to the fabrication of a layered carbon nitride membrane with interlayer 2D pores. The incomplete polymerization leads to the presence of terminal –NH groups in the carbon nitride nanosheets, which induces a negative charge on the membrane in a hydrated state. Owing to the cation selectivity, a 250 nm-thick free-standing carbon nitride membrane can generate a 0.21 Wm^−2^ maximum output power from a 1000-fold salinity gradient of KCl [88]. The metal carbide nitride nanosheets, widely known as MXene, are an emerging building block for ion-selective 2D membrane fabrication. The general formula of MXene is M_n+1_X_n_T_x_, where M is a transition metal and X is carbon and/or nitride. The synthesis of MXene involves the selective etching of the A-group layer from M_n+1_AX_n_. During the etching and exfoliation, the surface terminal groups (T_x_: -O, -OH and –F) replace the A layer. These functional groups induce surface charges in the MXene nanosheets and dictate the interlayer distance to some extent. The selective etching of the Al layer from Ti_3_AlC_2_ by hydrofluoric acid leads to the formation of Ti_3_C_2_T_x_, MXene. Like graphene oxide, vacuum-assisted filtration also allows for the fabrication of 2D lamellar Ti_3_C_2_T_x_, MXene membranes. At a 1000-fold salinity gradient of KCl, free standing Ti_3_C_2_T_x_, MXene membranes that are 3 µm in thickness can achieve a maximum output power density of 21 Wm^−2^ [89]. The Ti_3_C_2_T_x_, MXene nanosheets typically display a negative surface charge (−2.2 mCm^−2^) in a hydrated state. Therefore, it has a tendency to adsorb positively charged polyelectrolytes [90]. MXenes with a positive surface charge have been prepared by adding positively charged polyelectrolytes in the colloidal dispersion of Ti_3_C_2_T_x_, MXene nanosheets, e.g., polydiallyl dimethyl ammonium, polyethylene imine, polyallylamine hydrochloride, etc. [90]. The degree of the protonation follows the order- polydiallyl dimethyl ammonium > polyethylene imine > polyallylamine hydrochloride. Upon the functionalization of Ti_3_C_2_T_x_, MXene nanosheets with polydiallyl dimethyl ammonium, the surface charge density becomes +1.6 mCm^−2^, which, in the case of polyallylamine hydrochloride, is +0.5 mCm^−2^. The sequential vacuum filtration of positively and negatively charged MXene nanosheets leads to the formation of a diode-type free-standing membrane. A diode-type membrane containing negative Ti_3_C_2_T_x_, MXene nanosheets and polydiallyl dimethyl ammonium-adsorbed positive MXene nanosheets shows ion current rectification. The membrane produces an 8.6 Wm^−2^ maximum output power density at a 50-fold salinity gradient of NaCl, while, for a 500-fold salinity gradient, the output power density reaches 17.8 Wm^−2^ [90]. A strategy for enhancing the output power of the 2D membranes is to use a charged intercalating agent between the nanosheets. An aramid nanofiber derived from commercial Kevlar has been utilized as an intercalating agent between the Mxene nanosheets. The dispersion of Ti_3_C_2_T_x_, MXene and aramid nanofibers in dimethylsulfoxide shows a strong Tyndall effect. To prepare a free-standing composite membrane, the dispersion is vacuum-filtered on top of an anodized aluminum filter and peeled off from the substrate. The resulting mechanically strong free-standing composite membrane with a 2D ion transport channel demonstrates a 3.7 Wm^−2^ output power density for a 50-fold salinity gradient of NaCl, i.e., artificial seawater and river water [91].

### 4.5. Nanofiber-Based 3D Porous Membranes

Nanofibers are utilized as building blocks to fabricate membranes with 3D interconnected pores. The mechanical robustness of this class of membranes is governed by the surface interactions between the nanofibers, e.g., the hydrogen bond, van der Waals forces, etc. Free-standing membranes with sufficient mechanical properties are fabricated by both blade coating and vacuum filtration. The most challenging task of fabricating this class of membranes is controlling the space-charge effect [19]. Although nanofibers with a condensed charge at the surface can be synthesized by several methods, neither blade coating nor vacuum filtration allow for any control over the assembly of the nanofibers. The 3D interconnected pores in this class of membranes result from the completely random deposition of nanofibers. Several nanofibers-based porous membranes show an excellent power density in spite of this limitation. The Kevlar yarns have carved out a reputation for excellent mechanical properties, which allow for its application in body armors. Recently, it has been demonstrated that the Kevlar yarns can be split into nanofibers that are 5–30 nm in diameter [91,92,93,94]. The commercial Kevlar fiber is composed of aligned poly(paraphenylene terephthalamide). By the abstraction of the proton from the poly(paraphenylene terephthalamide) using saturated potassium hydroxide in dimethylsulfoxide, the Kevlar fiber can be chemically split into aramid nanofibers. Free-standing aramid nanofiber membranes with 3D interconnected pores are prepared by both blade coating and vacuum filtration. The oxygen-containing groups (e.g., carboxylic, hydroxyl) at the surface induce a negative charge to the nanofibers. When the membrane is placed between salt solutions of two different concentrations, diffusion current is generated, owing to the selective cation transport from the high-concentration side to the low-concentration side [92]. The maximum output power density is 4.8 Wm^−2^ for a 50-fold salinity gradient of NaCl, while it reaches 15 Wm^−2^ for a 500-fold salinity gradient of NaCl [92]. A heterogeneous membrane containing one layer of aramid nanofiber and one layer of polyelectrolyte hydrogel produced a maximum output power density of 5.06 Wm^−2^ for a 50-fold salinity gradient of NaCl [95]. Porous membranes containing biomass-derived nanowires have also been utilized for osmotic power generation. A biomass-derived nanowire fabrication technique is reported, using the commercial poly(ethylene oxide)-*b*-poly(propylene oxide)-*b*-poly(ethylene oxide) triblock copolymer Pluronic F127 as a structure-directing agent. Ribose is used as the carbon source to prepare the nanowire, while poly(4-styrenesulfonic acid-co-maleic acid) sodium salt and 1,3,5-trimethylbenze are used as structure-directing agents. The nanowires are deposited on an anodic alumina oxide layer by vacuum filtration to prepare the membrane, which produces a maximum output power density of 2.78 Wm^−2^ for a 50-fold salinity gradient of NaCl [96]. Vacuum-assisted filtration allows for the deposition of silk nanofibers from its dispersion in water on anodic aluminum oxide membranes [26]. In order to obtain silk nanofibrils from bombyx mori (silkworm), it is necessary to remove the sericin proteins by solvent treatment to deconstruct the hierarchical structure of silk fibers. A boiled aqueous solution of sodium bicarbonate is used to degum the nanofibrils by removing sericin proteins. Silk nanofibrils that are ~20 nm in diameter are obtained after incubation using 1,1,1,3,3,3-hefluro-2-propanol at 60 °C, followed by centrifugation. Owing to the presence of hydroxyl, carboxyl and amino groups at the surface of the silk nanofibrils, a hydrogen bond is formed between the anodic aluminum oxide and silk nanofibril layer, resulting in a hybrid membrane with excellent water stability. A maximum power density of 2.86 Wm^−2^ is obtained for a 50-fold salinity gradient of NaCl using a double-layer membrane with a 5 µm-thick silk nanofibril layer on a 60 µm-thick anodic aluminum oxide layer [26].

### 4.6. Metal Organic Frameworks (MOF) Containing Membranes

Metal organic frameworks (MOF) have received a lot of attention for several membrane-based applications. UiO-66 (UiO stands for the University of Oslo) is a zirconium-based porous MOF with 12 benzene-1,4-dicarboxylate linkers [97]. The voltage-driven ion transport (i.e., ion current rectification) through the single-porous polyethylene terephthalate membrane was successfully regulated by growing UiO-66 in situ at the interior of the channel [98]. UiO-66 crystals containing amino-substituted ligands (UiO-66-NH_2_) were grown on top of a porous alumina layer to prepare a heterogeneous membrane for osmotic power generation. The procedure involves the surface modification of the alumina layer using 3-aminopropyltriethoxysilane (APTES), followed by the in situ solvothermal synthesis of a UiO-66-NH_2_ layer. The APTES layer acts as a covalent linker between the alumina support and the UiO-66-NH_2_ layer [99]. Owing to the –NH_2_ functional groups, the UiO-66-NH_2_ is positively charged, which allows for the selective transport of anions. Under 5-, 50- and 500-fold KCl gradients, the maximum output power densities are 2.19, 4.93 and 7.12 Wm^−2^, respectively. The UiO-66-NH_2_ contains 6-7 Å-sized apertures. Several hydrated anions have sizes larger than the size of the apertures of UiO-66-NH_2_. Hence, if the inherent sub-nanometer pores of the UiO-66-NH_2_ determine the ion selectivity, only the anions smaller than the pores would pass through the membrane, and the larger hydrated anions would be rejected. However, the sequence of the anion transport through the membrane was Br^−^ (6.62 Å) < Cl^−^ (6.64 Å) < NO^3−^ (6.7 Å) < SO_4_^2−^ (7.58 Å). Therefore, the selective anion transport through the membrane is largely dictated by the adsorption affinity of the anions with the positively charged UiO-66-NH_2_ layer, and the inherent sub-nanometer pores of the UiO-66-NH_2_ do not contribute to the selectivity. A so-called “Ion Pool” membrane is reported by sandwiching an anodic aluminum oxide (AAO) membrane between a thin tungsten oxide (WO_3_) and a ZIF-8 layer (WO_3_-AAO-ZIF-8). The tungsten oxide layer is deposited on one side of the AAO membrane by reactive direct-current magnetron sputtering, followed by annealing at 500 °C to crystallize WO_3_. The resulting WO_3_ layer consists of intergranular gaps that allow for the permeation of ions. Spin coating is utilized to coat a polyvinyl alcohol solution containing ZIF-8 powder on the other side of the AAO membrane. The pristine AAO membrane with pores that are 25 nm in diameter do not show any ion current rectification behavior. However, the double-layered membranes containing a WO_3_ layer on an AAO layer (AAO-WO_3_) and a ZIF-8 layer on an AAO layer (AAO-ZIF-8) show ion current rectification. The geometrical and surface charge asymmetry of the AAO-WO_3_ and AAO-ZIF-8 membranes induces ion current rectification. The triple-layered WO_3_-AAO-ZIF-8 membrane shows stronger ion current rectification compared to the double-layered AAO-WO_3_ and AAO-ZIF-8 membranes. Using a 50-fold salinity gradient of NaCl, a 1.93 Wm^−2^ output power was obtained by the WO_3_-AAO-ZIF-8 membrane. However, it is noteworthy that this power density was obtained when the pH of the ZIF-8 side was adjusted to 3 and the pH of the WO_3_ side was adjusted to 11. The ZIF-8 layer is positively charged, while the WO_3_ layer is negatively charged at pH 11 [100].

### 4.7. Membranes Containing Mesoporous Carbon and a Silica Layer

Mesoporous carbon- and silica-based materials are reputed for their long-range-ordered interconnected pores and high surface area. The formation of this class of porous materials typically involves a crosslinkable precursor (as a source of carbon or silica) and a block copolymer as a templating agent. A nanoscale-ordered feature is produced by block copolymer self-assembly, followed by crosslinking and calcination. The precursor must have a favorable interaction with one of the blocks of the polymer, which will contribute to the formation of the majority phase of the ordered structure. The majority phase contributes to the formation of the continuous matrix (i.e., the pore wall) of the porous structure. The other block of the polymer forms a minority phase of the ordered structure, which eventually contributes to the formation of ordered pores. Upon thermal treatment, the crosslinking of the precursor takes place, while at a higher temperature, calcination occurs, and the template decomposes completely. The overall volume fractions of the minority and majority phase originating from the templating agent and a precursor dictate the morphology of the mesoporous structure. Hence, it is essential to choose a precursor that will lead to the final inorganic porous structure having a sufficient mechanical strength. Several combinations of block copolymers and precursors have been successfully utilized to produce ordered mesoporous structures [101]. The polyphenolic resin resol is a popular precursor for producing mesoporous carbon materials, while mesoporous silica is often produced using tetraethyl orthosilicate (TEOS) as a precursor. The commercial triblock copolymers of the Pluronic family composed of linear poly(ethylene oxide)-*block*-poly(propylene oxide)-*block*-poly(ethylene oxide) (PEO-*b*-PPO-*b*-PEO) are a popular templating agent. The co-assembly of Pluronic F127/resol [102] and Pluronic F127/TEOS [103] is widely used to produce mesoporous carbon and mesoporous silica, respectively. Moreover, the co-assembly of Pluronic F127/resol/TEOS [104] has been utilized to produce a hybrid carbon–silica mesoporous structure as well. An ionic diode membrane containing a negatively charged mesoporous carbon layer and a positively charged alumina layer shows cation selectivity. The membrane fabrication procedure involves filling up pores of the macroporous alumina support layer with poly(methyl methacrylate) before coating the precursor solution (containing Pluronic F127 and resol) on top. The evaporation-induced self-assembly of the precursor solution followed by calcination at 450 °C produces a highly ordered mesoprous carbon layer with a pore diameter of ~6.7 nm. The poly(methyl metharylate) degrades completely during the calcination. For a 50-fold salinity gradient of NaCl, a maximum power density of 3.46 Wm^−2^ is obtained [105]. An ordered silica mesoporous layer on an anodic alumina support has also been used for NRED/NPG. The membrane is obtained by the evaporation-induced assembly of Pluronic F127/TEOS on top of a poly(methyl methacrylate)-filled anodic alumina layer, followed by crosslinking TEOS at 100 °C and calcination at 500 °C. The diode-type membrane containing a negatively charged mesoporous silica layer on the positively charged anodic alumina oxide layer shows a 4.5 Wm^−2^ maximum power density for a 50-fold salinity gradient of NaCl [103]. Using the same technique, poly(methyl methacrylate)-filled anodic alumina, a hybrid carbon-silica mesoporous layer is fabricated by the evaporation-induced self-assembly of Pluronic F127/resol/TEOS. The resulting membrane is composed of a negatively charged carbon–silica mesoporous layer with a 6 nm pore size on top of a positively charged anodic alumina layer with 80 nm pores. While a 5.04 Wm^−2^ maximum power density is obtained for a 50-fold salinity gradient of NaCl using the membrane, for a 200-fold salinity gradient, the maximum power density reached 10.75 Wm^−2^ [106].

### 4.8. Porous Block Copolymer Membranes

The self-assembly of a block copolymer is not only utilized to prepare sacrificial templates for membrane fabrication (as described in Section 4.7), but the porous membrane itself can also be prepared by taking advantage of the self-assembly of the block copolymers. Isoporous block copolymer membranes [107,108,109] have already carved out a reputation for other membrane-based applications, e.g., ultrafiltration, nanofiltration, etc. For osmotic power generation, the potential of block copolymer membranes remains largely unexplored. A double-layer porous membrane containing a spin-coated polystyrene-*block*-poly(4vinylpyridine) (PS-*b*-P4VP) layer on top of a track-etched poly(ethylene terephthalate) layer with conical pores has been reported (Figure 5). The vinyl pyridine moieties of the spin-coated layer are protonated under acidic pH and acquire a positive charge. Under this condition, the pore-forming block (i.e., protonated P4VP) acquires a stretched conformation, and the thin layer of the membrane contains positively charged soft nanochannels. At pH 4.3, the double-layer membrane acts as an anion-selective membrane. The membrane produces a 0.35 Wm^−2^ maximum power density for a 50-fold salinity gradient of NaCl at pH 4.3 [110]. A Janus-type membrane with a porous PS-*b*-P4VP layer and a porous crosslinked block copolymer substrate produced a 2.04 Wm^−2^ maximum power density owing to the mixing of electrolytes containing a 50-fold salinity gradient of NaCl at pH 4.3 [111]. An amphiphilic liquid crystalline block copolymer containing a poly(ethyelene oxide) minor block is used to prepare the substrate layer. The major block of the block copolymer is composed of photocrosslinkable chalcone-functionalized methacrylate repeating units. The major block and the minor block of the polymer are linked with a photocleavable o-nitrobenzyl ester linker. The spin coating of the polymer solution followed by thermal annealing and UV light treatment leads to a crosslinked substrate with carboxylic moieties along the pore wall. Spin coating a PS-*b*-P4VP monomer on top of this substrate leads to the formation of a Janus membrane, as the minor P4VP can acquire a positive charge under acidic conditions. The resulting membrane selectively transports anions. While the concept can be used to prepare Janus membranes, it is essential to ensure the alignment of the positively charged and negatively charged pores to maximize the power conversion (Figure 5). For block copolymer porous membranes, both the pore size and pore-to-pore distance can be tuned by varying the molecular weight and composition of the block copolymer. Since both of the features are of prime importance for selective nanofluidic ion transport through the block copolymer membrane, this class of membrane has a huge potential for NRED/NPG.

### 4.9. Other Porous Membranes

The major concepts of porous membrane development for NRED/NPG are summarized in Section 4.1, Section 4.2, Section 4.3, Section 4.4, Section 4.5, Section 4.6, Section 4.7 and Section 4.8. Apart from these, other membrane fabrication techniques for NRED/NPG have been explored as well. Utilizing the surface charge density within the tandem gaps of two multi-porous membranes is an interesting approach to improving the ion selectivity. The tandem gap confinement up to 500 nm was constructed by mechanically extruding two 1D track-etched PET membranes or two 3D porous cellulose acetate membranes. A 4.72 Wm^−2^ output power density is obtained by stacking multiple cellulose acetate membranes [112]. Recently, it has been demonstrated that filling up the pores of an anodic alumina oxide membrane with polyvinyl alcohol gel increases the ion transport under the condition where the EDL overlaps in the pore. The maximum power density increased twofold owing to the introduction of polyvinyl alcohol in the pore [113]. A hybrid membrane has been fabricated by synthesizing COF-LZU1 (Lan Zhou University1) on a cellulose nanofiber support using a carbon nanotube intermediate layer between the COF-LZU1 and the cellulose nanofiber layer. The COF-LZU1 is prepared from a reaction between 1,3,5-Benzenetricarboxaldehyde and 1,4-diaminobenzene. A 4.26 Wm^−2^ maximum power density is obtained from a 50-fold salinity gradient of NaCl using the hybrid membrane [114].

## 5. Summary and Outlook

In the last two decades, substantial progress occurred in the development of RED stack components and the optimization of an operational condition, which led to the enhancement of the achievable net power from an RED stack [23]. RED went beyond the laboratory-scale units and has been implemented in pilot plants due to the progress of the technology [115,116]. The first pilot-scale prototype of an RED stack for generating electricity from brine was commissioned in March 2014 using membranes from Fujifilm Manufacturing Europe BV [117]. Until now, such large-scale development of RED was strictly limited to conventional non-porous membranes. The utilization of porous membranes for NRED/NPG is a relatively new field. In this review, the recent progress of the membrane development for NRED/NPG has been thoroughly reviewed (Section 4). The maximum power densities obtained by the multi-porous membranes are tabulated in Table 1, along with the salinity gradient used to perform the experiments. The single-porous membranes are not included in Table 1, as the power density of such membranes is often estimated by linear extrapolation at a high pore density. It is well established that such linear extrapolation is valid only up to a certain pore density (mentioned in Section 1). Table 1 shows that there are several multi-porous membranes that show an excellent maximum power density. While there have been major developments in the membrane design concepts of the laboratory-scale investigation of NRED/NP, the large-scale application remains uncertain. Extensive development is required for the promulgation and practical implementation of this class of membrane. The commercial benchmark of maximum power density is 4–6 Wm^−2^. Several porous membranes having potentials of scaling up have surpassed this benchmark in laboratory investigation using a working area below 0.1 mm^2^ [118]. It is extremely important to use a larger working area in order to estimate a practically attainable combination of energy efficiency and maximum output power density. While the maximum power density is a useful parameter for figuring out the potential of a membrane fabrication technique, the gross power of the RED stack is dependent on several additional factors (briefly discussed in Section 2). Unlike the conventional non-porous ion-exchange membranes, the current status of the porous membrane development does not allow for an analysis beyond the mere comparison of the maximum power densities of the membranes. Moreover, so far, the development of porous membranes for NRED/NPG has been largely limited to the fabrication of cation-selective membranes, i.e., membranes with negatively charged surfaces. In principle, it is possible to stack only the cation-selective membranes in series to extract energy [119]. However, this will make only half of the Gibbs free energy available to be converted into electrical energy. This is because, in such a stack, the ion current migration across the membrane will not take place due to the mixing of anions. To explore the full potential of converting Gibbs free energy into extractable electrical energy, it is imperative to use the alternating series of anion and cation exchange membranes in an NRED/NPG stack. Hence, the development of porous anion exchange membranes is equally important to the development of cation exchange membranes. However, the anion exchange porous membrane development for NRED/NPG unfortunately remains largely unexplored so far. The influence of the variation of the salinity gradient is often investigated during nanofluidic porous membrane development using electrolytes containing KCl and NaCl. A 50-fold salinity gradient of NaCl is typically used to mimic the mixing of seawater and river water. It has been known for decades that seawater contains a considerable amount of divalent salt. The presence of divalent ions generally reduces the maximum power density of the membrane owing to their uphill transport from the low-concentration side to the high-concentration side as a response of the developed electrical potential across the ion selective membrane [24]. Most of the studies focused on the development of NRED/NPG membranes completely ignore the negative impact of divalent salts in the laboratory-scale experiments. For the recently developed MTPP-COF monolayer membranes [82], it has been demonstrated that the presence of divalent ions in the electrolyte solution increases the output power density. To mimic the mixing of seawater and river water, electrolyte solutions containing a mixture of NaCl and divalent ion (Mg^2+^, Ca^2+^ and SO_4_^2−^)-containing salt have been used. An unprecedented output power density of 203.8 Wm^−2^ is obtained for a 50-fold salinity gradient. Moreover, real seawater (from China’s Yellow Sea) and river water (from Yangtze River) have been used to demonstrate that the MTPP-COF monolayer membrane produces a 294.3 Wm^−2^ output power initially, which reduces to 250 Wm^−2^ in four days and remains rather stable up to 7 days [82]. The outstanding output power density and the time-dependent investigation of the output power density using real seawater and river water in this study serve as a role model for future laboratory experiments, paving the way for the real application of NRED/NPG.

## Figures and Tables

**Figure 1 membranes-13-00164-f001:**
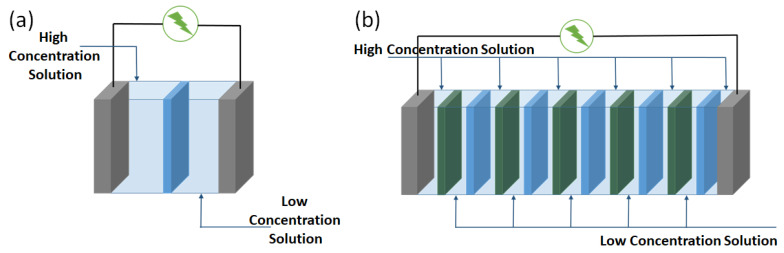
Schematic representation of (**a**) the half-cell system, (**b**) the RED stack, i.e., the full-cell system.

**Figure 2 membranes-13-00164-f002:**
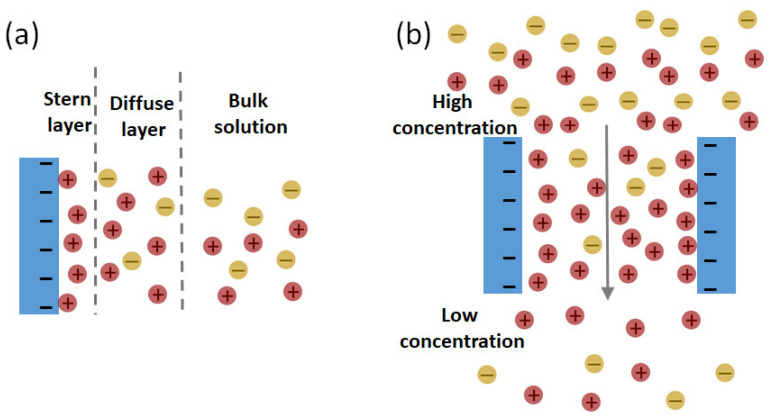
Schematic representation of (**a**) an electrical double layer at a negatively charged surface, (**b**) an overlapped electrical double layer inside a negatively charged pore.

**Figure 3 membranes-13-00164-f003:**
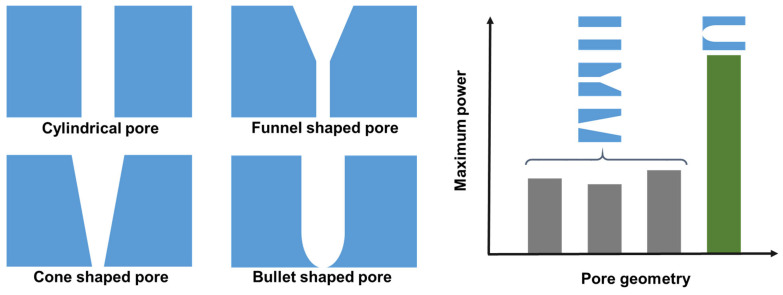
Schematic representation of different pore geometries in track-etched membranes. The bullet-shaped pore generates higher power compared to the pores of other geometries.

**Figure 4 membranes-13-00164-f004:**
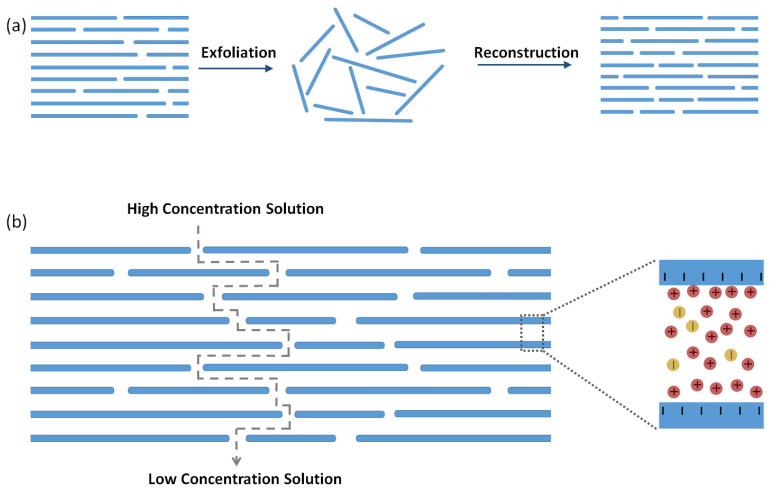
Schematic representation of (**a**) the exfoliation–reconstruction approach to fabricating layered membranes with 2D pores, (**b**) the selective ion transport through 2D pores due to the overlap of the electric double layer at the interlamellar gap of the nanosheets.

**Figure 5 membranes-13-00164-f005:**
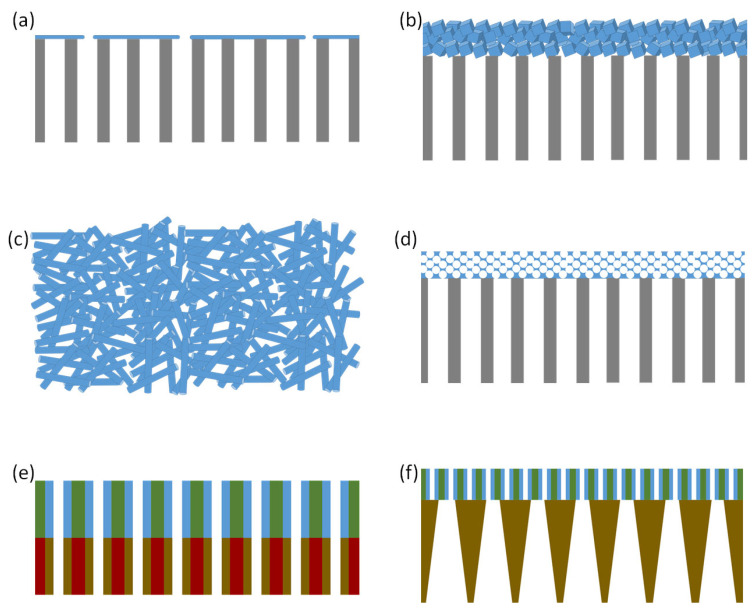
Schematic representation of membranes with (**a**) an ultrathin porous layer with atomic- or molecular-scale thickness on a porous support, (**b**) a metal organic framework layer on a porous support. (**c**) randomly assembled nanofibers with 3D interconnected pores, (**d**) a mesoporous carbon or silica layer on porous support, (**e**) two porous block copolymer layers, (**f**) a porous block copolymer layer on top of a track-etched layer.

**Table 1 membranes-13-00164-t001:** Comparison of the maximum power densities obtained from different types of multi-porous membranes.

Membrane Type	Membrane Description	Concentration Gradient	Maximum Power Density	Reference
Atomic and molecularly thin porous membranes	Multi-porous hexagonal boron nitride membrane	1000-fold (KCl)	15 Wm^−2^	[77]
	Multi-porous graphene sheets containing –NH_2_ groups at the pore edges	100-fold (KCl)	35 Wm^−2^	[80]
	Crosslinked core–rim structure polycyclic aromatic hydrocarbon monomer hexa(2,2′-dipyridylamino)hexabenzocoronene	50-fold (NaCl)	67 Wm^−2^	[81]
	metal tetraphenylporphyrin COF (MTPP-COF) monolayer	50-fold (NaCl) 50-fold (MgCl_2_) 50-fold (CaCl_2_)	135.8 Wm^−2^ 317.5 Wm^−2^ 267.7 Wm^−2^	[82]
Nanofluidic membranes with 2D pores	Layered carbon nitride membrane	1000-fold (KCl)	0.21 Wm^−2^	[88]
	Free-standing Ti_3_C_2_T_x_, MXene membrane	1000-fold (KCl)	21 Wm^−2^	[89]
	Diode-type membrane containing negative Ti_3_C_2_T_x_, MXene nanosheets and polydiallyl dimethyl ammonium-adsorbed positive MXene nanosheets	50-fold (NaCl) 500-fold (NaCl)	8.6 Wm^−2^ 17.8 Wm^−2^	[90]
	Aramid nanofiber intercalated Ti_3_C_2_T_x_, MXene nanosheets	50-fold (NaCl)	3.7 Wm^−2^	[91]
Nanofiber-based 3D porous membranes	Free-standing aramid nanofiber membrane	50-fold (NaCl) 500-fold (NaCl)	4.8 Wm^−2^ 15 Wm^−2^	[92]
	Double-layer membrane containing one layer of aramid nanofiber and one layer of polyelectrolyte hydrogel	50-fold (NaCl)	5.06 Wm^−2^	[95]
	Double-layer membrane containing nanowires deposited on a porous anodic alumina oxide layer	50-fold (NaCl)	2.78 Wm^−2^	[96]
	Double-layer membrane containing a silk nanofibril layer and a porous anodic aluminum oxide layer	50-fold (NaCl)	2.86 Wm^−2 ^	[26]
Metal organic frameworks (MOF) containing membranes	Double-layer membrane containing an amino-substituted UiO-66 layer on a porous alumina layer	5-fold (KCl) 50-fold (KCl) 500-fold (KCl)	2.19 Wm^−2^ 4.93 Wm^−2^ 7.12 Wm^−2^	[99]
	“Ion Pool” membrane containing a sandwiched anodic aluminum oxide (AAO) layer between a tungsten oxide (WO_3_) layer and a ZIF-8 layer (WO_3_-AAO-ZIF-8)	50-fold (NaCl)	1.93 Wm^−2^	[100]
Membranes containing a mesoporous carbon and silica layer	Double-layer membrane with a mesoporous carbon layer on a porous alumina layer	50-fold (NaCl)	3.46 Wm^−2^	[105]
	Double-layer membrane with a mesoporous silica layer on a porous alumina layer	50-fold (NaCl)	4.5 Wm^−2^	[103]
	Double-layer membrane with a mesoporous carbon–silica hybrid layer on a porous alumina layer	50-fold (NaCl) 200-fold (NaCl)	5.04 Wm^−2^ 10.75 Wm^−2^	[106]
Porous block copolymer membranes	Double-layer membrane containing a spin-coated polystyrene–*block*–poly (4vinylpyridine) (PS-*b*-P4VP) layer on top of a track-etched poly(ethylene terephthalate) layer	50-fold (NaCl)	0.35 Wm^−2^	[110]
	Janus type membrane having a porous PS-*b*-P4VP layer and a porous crosslinked block copolymer substrate containing a poly (ethyelene oxide) minor block	50-fold (NaCl)	2.04 Wm^−2^	[111]
Covalent organic framework (COF)-containing membrane	Hybrid membrane with COF-LZU1 on a cellulose nanofiber support with a carbon nanotube intermediate layer	50-fold (NaCl)	4.26 Wm^−2^	[114]

## Data Availability

Not applicable.

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
