# Peer review of "Membranes for Osmotic Power Generation by Reverse Electrodialysis"

_membranes, 2023, doi:10.3390/membranes13020164_

Round 1

Reviewer 1 Report

This is an excellent review of an emerging topic and extensively covers applied use of reverse electrodialysis and covers membrane fabrication and strategies for producing ion selectivity as well as covering nascent nanofluidic reverse electrodialysis for optimized power generation in RED.

Author Response

The manuscript has been throughly revised to correct typing mistakes and spelling errors.

Reviewer 2 Report

1-    It is necessary to describe the mass transfer mechanism for each kind of membranes.

2-    It is necessary to quantitatively express the results of using any type of membrane in electricity production.

3-    It is suggested to give a comparison regarding the performance of different membranes in osmotic power generation by RED.

4-      The Summary and Outlook can be written better and more completely so that the prospect of using different membranes in power generation is fully defined.

5-    Discuss the advantages and disadvantages of using each type of membrane in power generation by RED

Author Response

1-    It is necessary to describe the mass transfer mechanism for each kind of membranes.

Author’s response – The mass transfer through nonporous membrane is discussed in section 3 and porous membrane is discussed in section 4.1 with sufficient details.

2-    It is necessary to quantitatively express the results of using any type of membrane in electricity production.

Author’s response – In response to the suggestion of the reviewer a comparison of the maximum power density of different type of membranes are provided in table 1.

3-    It is suggested to give a comparison regarding the performance of different membranes in osmotic power generation by RED.

Author’s response – In response to the suggestion of the reviewer a comparison of the maximum power density of different type of membranes are provided in table 1. The maximum power density is widely used to compare the performance of the porous membranes.

4-      The Summary and Outlook can be written better and more completely so that the prospect of using different membranes in power generation is fully defined.

Author’s response – If the reviewer thinks he could have written a better summary and outlook I do not want to make any comment to disrespect the reviewer. But in my opinion, nanofluidic reverse electrodialysis as a research field did not make enough progress so that  one can analyze the prospect of using different membranes in power generation. In the revised version of the article I added some statements to clarify the point.

“Utilization of porous membranes for NRED/NPG is a realtively new field. In this review the recent progress of the membrane development for NRED/NPG have been thoroughly reviewed (Section 4). The maximum power density of the obtained by the multiparous membranes are tabulated in Table 1 along with the salinity gradient used to perform the experiments. The single porous membranes are not included in table 1 as the power den-sity of such membranes are often estimated by linear extrapolation at a high pore density. It is well established that such linear extrapolation is valid only up to a certain pore den-sity (mentioned in section 1). Table 1 shows there are several multiporous membranes which shows excellent maximum power density.    While there have been major de-velopment in the membrane design concepts laboratory scale investigation of NRED/NP, the large scale application remains uncertain. Extensive development is required for the promulgation and practical implementation of this class of membrane. The commercial benchmark of maximum power density is 4 – 6 Wm-2. Several porous membranes having potentials of scaling up have surpassed this benchmark in laboratory investigation using working area below 0.1 mm2.[118] It is extremely important to use larger working area in order to estimate a practically attainable combination of energy efficiency and maximum output power density. While a the maximum power density is a useful parameter to figure out the potential of an membrane fabrication technique the gross power of the RED stack is depended on several additional factors (briefly discussed in section 2). Unlike the con-ventional nonporous ion-exchange membranes, the current status of the porous mem-brane development does not allow an analysis beyond mere comparison of maximum power density of the membranes.”

5-    Discuss the advantages and disadvantages of using each type of membrane in power generation by RED

Author’s response – The advantage and disadvantage of using different types of membranes are discussed in the paper to some extent in section 4. Nanofluidic reverse electrodialysis membranes is a new field of research. In my opinion a more detail discussion about the advantage and disadvantage of each type of membrane is not possible.